# LargePiG for Hallucination-Free Query Generation: Your Large Language Model is Secretly a Pointer Generator

## ABSTRACT

Recent research on query generation has focused on using Large Language Models (LLMs), which despite bringing state-of-the-art performance, also introduce issues with hallucinations in the generated queries. In this work, we introduce relevance hallucination and factuality hallucination as a new typology for hallucination problems brought by query generation based on LLMs. We propose an effective way to separate content from form in LLM-generated queries, which preserves the factual knowledge extracted and integrated from the inputs and compiles the syntactic structure, including function words, using the powerful linguistic capabilities of the LLM. Specifically, we introduce a model-agnostic and training-free method that turns the **Large** Language Model into a **P**ointer-**G**enerator (**LargePiG**), where the pointer attention distribution leverages the LLM's inherent attention weights, and the copy probability is derived from the difference between the vocabulary distribution of the model's high layers and the last layer. To validate the effectiveness of LargePiG, we constructed two datasets for assessing the hallucination problems in query generation, covering both document and video scenarios. Empirical studies on various LLMs demonstrated the superiority of LargePiG on both datasets. Additional experiments also verified that LargePiG could reduce hallucination in large vision language models and improve the accuracy of document-based question-answering and factuality evaluation tasks. The source code and dataset are available at https://anonymous.4open.science/r/LargePiG-7674.

## KEYWORDS

Query Generation, Hallucination, Pointer Generator

**ACM Reference Format:**
Anonymous Author(s). 2024. LargePiG for Hallucination-Free Query Generation: Your Large Language Model is Secretly a Pointer Generator. In . ACM, New York, NY, USA, 19 pages. https://doi.org/10.1145/nnnnnnn.nnnnnnn

## 1 INTRODUCTION

Query generation is an automatic process of generating queries according to the content presented in documents or videos, which not only facilitates information retrieval from documents [12, 35, 48] but also serves applications like short video platforms by creating queries that attract user engagements. There has been notable advancement in query generation using LLMs [5, 12, 36, 39]. However,

employing LLMs for query generation often introduces hallucination issues. **Factuality hallucination** refers to inaccuracies in the facts presented in the generated queries, often occurring when the inputs include knowledge not covered by the LLM's pre-training data. For example, being misled by the latest facts in the news documents can make LLMs generate queries that conflict with actual events. **Relevance hallucination** occurs when the generated queries, although factually correct, are irrelevant to the inputs [15]. Both types of hallucinations are not mutually exclusive, with some generated queries exhibiting both issues (see appendix A.1 for the experimental validation of hallucination classification).

Previous research has primarily focused on reducing relevance hallucinations through post-processing methods [5, 12, 15], without addressing hallucinations at the source of generation. With the expanding range of applications for query generation on short-video platforms, generating "related search" based on video content to attract user clicks and enhance user engagement has become crucial for these platforms. Figure 1 presents some examples of "related search" on short-video platforms, each of which has hundreds of millions of users [1]. If a generated query exhibits relevance hallucinations, users may not click the query as clicking on "related search" will not find content related to the video, diminishing user interest. Conversely, if a query demonstrates factuality hallucinations (without relevance hallucinations), it might initially attract users' interest through clickbait but fail to deliver content related to the hallucinatory facts, thereby degrading the user experience. Therefore, the queries we generate need to be relevant to the video content, factually accurate, and sufficiently novel to attract user clicks and improve user engagement.

Unlike other generation tasks, query generation primarily relies on the inputs. Thus, decoupling the content and form at the output end of LLMs, ensuring that the factual content of the generated queries mainly comes from the inputs and that the syntax and other forms are organized by LLMs, is key to keeping the generated query truthful and reducing hallucination issues. To this end, we propose to use the Pointer Generator (PG) technology, a sequence-to-sequence model that integrates extraction (pointing to words in the input) and generation (creating new words) strategies to enhance the accuracy and relevance of the generated text [42, 46]. The PG model, combines pointer attention distribution (determining the model's focus on different parts of the inputs), vocabulary distribution (the probability distribution for choosing the next word from a fixed vocabulary), and copy probability (deciding whether to generate a word from the vocabulary distribution or copy directly from the input), not only increases the probability of mentioning facts presented in the inputs and decreases the likelihood of generating unrelated facts but also ensures the correctness of syntax and other forms generated by LLMs. Although PG technology has

[1]TikTok: www.tiktok.com; Kwai: www.kuaishou.com; Xiaohongshu: www.xiaohongshu.com.

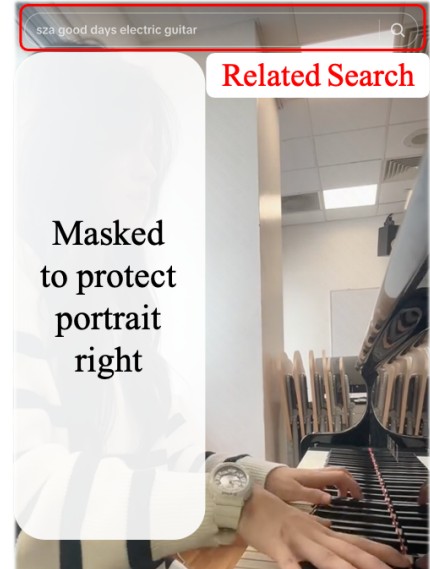 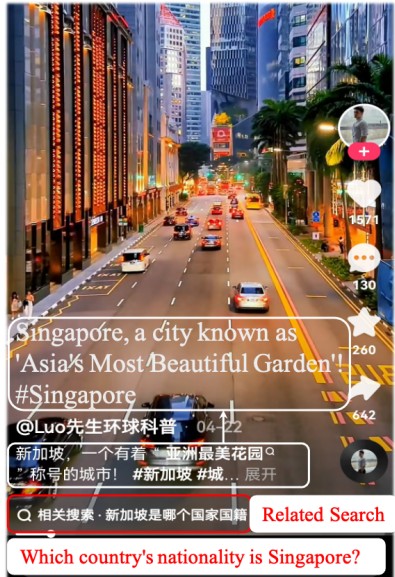 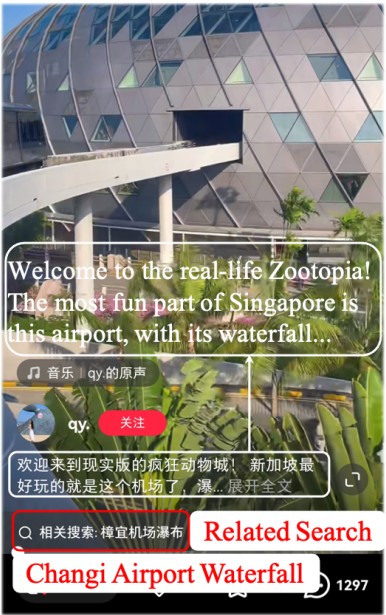

**(a)** Relevance Hallucination: The video is from TikTok, where the "related search" at the top presents a certain relevance hallucination, as the person in the video is playing an electric piano rather than an electric guitar.

**(b)** Factuality Hallucination: The video is from Kwai, where the "related search" presents a certain factuality hallucination. Singapore itself is a country, so it is illogical to ask which country's nationality it belongs to.

**(c)** Truthful Query: The video is from Xiaohongshu, where the "related search" at the top present is relevant and factual.

**Figure 1: Examples of query generation in real applications across different short video platform.**

been applied in query generation tasks with traditional language models [19, 47], considering the enormous parameter size and training resource consumption of LLMs, adopting the traditional PG scheme, which requires learning pointer attention distribution and copy probability, may not only disrupt the original representations of LLMs but also diminish their generalization capability.

Facing the above challenge, we propose a novel PG implementation that can achieve PG functionality within LLMs without requiring additional training. Our method is based on two core observations: (1) Attention modules are more 'truthful' than other modules in LLMs (e.g., FFN modules), allowing the intrinsic attention weights towards the input sequence within LLMs to serve as the PG's pointer attention distribution; (2) LLMs generate different types of words (function words and factual knowledge words) with distinct patterns [10, 41]. When generating function words, the vocabulary distribution obtained from the high layers of LLMs is relatively consistent, whereas, for factual knowledge words, the vocabulary distribution from the high layers of LLMs shows significant differences. Further analyzing the internal mechanism behind the occurrence of different patterns in LLMs, we find that this pattern is rooted in the difference in the amount of information between function words and factual knowledge words in human linguistics. We relaxed the requirement for LLMs to generate the correct words, only needing them to identify the type of word to be generated and calculate the copy probability through the difference between the vocabulary distribution of the model's high layers and the last layer.

Based on this concept, we propose that **Large** Language Models can essentially act as an implicit **P**ointer-**G**enerator (**LargePiG** 🐷), better addressing the hallucination issues in query generation. Our method has several notable advantages: Firstly, it preserves LLMs' powerful capabilities and generalizability, as it does not require significant modifications to the model architecture or additional training. Secondly, by simplifying the implementation process of PG, our method reduces additional computational and resource requirements, making it more efficient and easy to implement. Lastly, this approach retains the advantages of PG, achieving decoupling of content and form at the output end of LLMs, making the generated content faithful to the inputs.

To better assess the capability of LargePiG in solving hallucination issues within query generation, we introduce TruthfulVQG and TruthfulDQG, two challenging Truthful Query Generation benchmarks gathered from video and document scenarios, respectively. Experiments on these datasets demonstrate that LargePiG is capable of increasing the factuality and relevance of various LLM-based query generation methods across different LLMs. More experiments on the LLaVA [24] family validate the effectiveness of LargePiG in addressing hallucination issues in query generation within multimodal scenarios. Further experiments on relevance testing and factuality evaluation demonstrate that LargePiG can individually address relevance hallucination and factuality hallucination. Efficiency analysis shows that LargePiG causes negligible latency in

the query generation process, proving the practical applicability of LargePiG.

We summarize the major contributions of this paper as follows:

(1) We identify the relevance and factuality hallucination issues in query generation, which are crucial for ensuring effective "related search" in short-video platforms.

(2) We propose **LargePiG**, a training-free, and model-agnostic decoding method that mitigates query generation hallucinations without modifying LLM architectures, ensuring ease of deployment.

(3) We introduce two truthful query generation benchmarks, **TruthfulVQG** and **TruthfulDQG**, and demonstrate through extensive experiments the effectiveness of LargePiG in reducing hallucinations while maintaining efficiency.

## 2 RELATED WORK

**Large language models based query generation.** Query generation is vital for improving information retrieval systems and user experience on short video platforms. Doc2Query [35] implements this concept using a sequence-to-sequence model for generating queries based on document contents. Advancing this, UDP [40] utilizes LLMs in a zero-shot setting to predict query likelihood from text passages. Building on this, PQGR [12] and InPars [5] introduce few-shot and contrastive example approaches, enhancing the contextual awareness of query generation. AQG [25] further develops LLM adaptability to query generation by employing LoRA [17] for fine-tuning with real user queries and context, alongside other parameter-efficient methods like soft-prompt tuning and adapters [36, 37]. Additionally, UDAPDR [39] explores efficiency by combining large and small models to generate and refine queries. Our work addresses hallucination in query generation, introducing LargePiG, a novel decoding method applicable to LLM-based query generation approaches to reduce relevance and factuality hallucination.

**Hallucination mitigation in large language models.** Large Language Models exhibit a critical tendency to produce hallucinations, resulting in content that is inconsistent with real-world facts or user inputs. Hallucination mitigation strategies can be data-driven, involving more refined filtering of pretraining data [28] or high-quality instruction-tuning datasets [52] to reduce the likelihood of LLMs learning hallucinatory knowledge. Alternatively, approaches from the input side, such as Retrieval Augmented Generation, utilize data to reduce LLM-generated hallucinations by grounding the model with an external knowledge base [14]. However, Retrieval Augmented Generation is not well-suited for tasks like query generation, as there is no explicit need for external retrieval content. Our LargePiG method focuses on reducing hallucination for the query generation task from the generation side, transforming the LLM into a pointer generator by leveraging intrinsic features of the LLM to separate content and form in LLM-generated queries. Unlike DoLa [10], which contrasts between transformer layers to correct the next word's probability, LargePiG derives the copy probability from the difference between the vocabulary distribution of the model's high layers and the last layer. Moreover, these hallucination mitigation methods are orthogonal to the LargePiG approach taken in this paper and could potentially be used in conjunction to mitigate hallucinations further.

## 3 METHOD

Current Large Language Models are fundamentally based on the Transformer decoder-only architecture. Initially, the input text is tokenized and transformed into numerical vectors by the embedding layer. Given a sequence of input tokens as $X = \{x_1, x_2, \ldots, x_{t-1}\}$, where the input tokens may include the instruction $I = \{x_1, \ldots, x_{m-1}\}$, the source document $D = \{x_m, \ldots, x_n\}$, and part of generated query $\widetilde{Q} = \{x_{n+1}, \ldots, x_{t-1}\}$, the embedding layer first converts these tokens into a series of vectors $H_0 = \{h_1^{(0)}, \ldots, h_{t-1}^{(0)}\}$. After passing through multiple Transformer Decoder Layers, $H_N$ is processed by a Classification Layer, usually composed of a layer of linear layers and softmax, mapping to the vocabulary distribution.

To address the hallucination issues present in LLM-based query generation, we propose to incorporate the mechanism of the Pointer-Generator to enhance the model's faithfulness to the factual knowledge contained within the source document $D$. The Pointer-Generator combines the original decoding vocabulary distribution $P_{\text{vocab}}$ of the LLM with the newly introduced pointer attention distribution $P_{\text{source}}$, the latter representing the probability distribution over the source document $D$. Furthermore, the Pointer-Generator includes a copy probability $p_{\text{copy}}$, which determines whether the model selects the next word from a predefined vocabulary or directly copies a word from the source document. We propose to use this mechanism to ensure that the factual content in the generated query mainly comes from $D$ and that the syntax and other forms are organized by LLMs, significantly reducing the occurrence of hallucinations.

Unlike previous approaches that required retraining the pointer-generator model to learn the pointer attention distribution and copy probability, we propose **LargePiG**, a plug-in and training-free method, to implement pointer-generator decoding within LLMs (see Figure 2). The pointer attention distribution can utilize the LLM's intrinsic attention weights towards the source document (§ 3.1); the vocabulary distribution comes from the output of the original LLM, ensuring the generative capability of the model (§ 3.2); and the copy probability is derived from the difference between the vocabulary distribution of the model's high layers and the last layer (§ 3.3). Finally, we delve into the rationality of why LargePiG can implicitly transform LLM into a pointer generator (§ 3.4).

### 3.1 LargePiG: Pointer Attention Distribution

The core module of Large Language Models consists of $N$ stacked Transformer layers. Each Transformer layer contains a self-attention module and feedforward neural networks (FFN) to process the embedded vectors, allowing the model to focus on the most relevant parts of the input dynamically. As the vectors in $H_0$ pass through each Transformer layer, they are successively transformed, with the output of the layer $j$ represented as $H_j$. In this process, taking the layer $j$ as an example, $H_{j-1}$, the output of the layer $(j-1)$, first passes through the $j$-th layer's self-attention module. Here, we take Multi-Head Attention (MHA) as an example, which can be easily generalized to Multi-Query Attention [43] and Grouped-Query Attention [2]:

$$\text{MHA} = \text{Concat}(\text{head}_1, \ldots, \text{head}_M)W^O, \quad (1)$$

$$\text{head}_i = A_i(H_{j-1}W_i^Q, H_{j-1}W_i^K, H_{j-1}W_i^V), \quad (2)$$

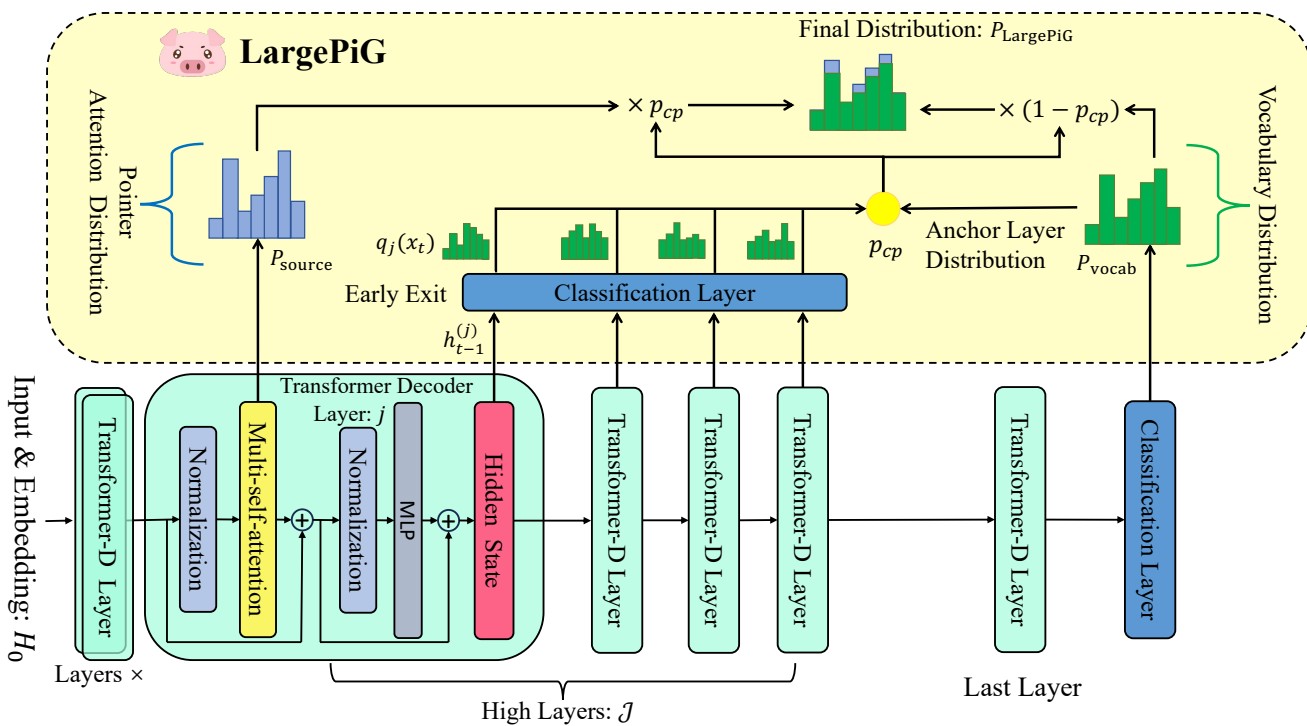

**Figure 2: The architecture of the proposed plug-in and training-free method LargePiG. Pointer Attention Distribution (§ 3.1) from the LLM's self-attention weights, Vocabulary Distribution (§ 3.2) from the output of the original LLM, Copy Probability (§ 3.3) from the difference between the vocabulary distribution of the model's high layers and the last layer.**

$$A_i(Q, K, V) = A_i^w V, \ A_i^w = \text{softmax}\left(\frac{QK^T}{\sqrt{d}}\right), \quad (3)$$

where $A_i^w$ denotes the attention weights of MHA, with $M$ as the number of heads, $W^{Q/K/V/O}$ are learnable parameters and $\sqrt{d}$ are scaling factor. Since each head captures a unique attention pattern, we aggregate these by averaging: $A^w = \frac{1}{M}\sum_{i=1}^{M} A_i^w$, enabling a unified representation of attention mechanisms across heads.

In the context of LargePiG, computing the pointer attention distribution $P_{\text{source}}$ primarily focuses on the attention weights from the last token in $H_{j-1}$ (i.e., $A_{t-1}^w$) to the tokens of the source document $D$. As the source document $D$ corresponds to tokens from $m$ to $n$ in the input sequence, we use $A_{t-1, m:n}^w$ to compute $P_{\text{source}}$. First, for the values in $A_{t-1, m:n}^w$, we normalize them to ensure their sum equals one, forming a probability distribution. Since we are only concerned with the tokens corresponding to the source document in $A^w$ and we already know this is extracted from a larger softmax function, direct normalization suffices. Let this normalized vector be $\mathbf{P}_{m:n}$:

$$\mathbf{P}_{m:n} = \frac{A_{t-1, m:n}^w}{\sum_{i=m}^{n} A_{t-1, i}^w} \quad (4)$$

Next, we construct the probability distribution to match the vocabulary distribution. We depart from traditional PG by not considering new word emergence, focusing on maintaining LLM generation fidelity to input while acknowledging the prevalent use of

sentence-piece tokenization [23]. Let $\mathcal{V}$ be the vocabulary of the LLM. The probability distribution for each token $x_i$ in $P_{\text{source}}$ within $\mathcal{V}$ comes from the corresponding attention weight in $\mathbf{P}_{m:n}$. Therefore, for each token $x_i$ in the vocabulary $\mathcal{V}$, its pointer attention distribution $P_{\text{source}}(x_i)$ is defined as:

$$P_{\text{source}}(x_i) + = \begin{cases} \mathbf{P}_{m:n}[j] & \text{for all } j \text{ where } x_j = x_i \text{ and } x_j \in D \\ 0 & \text{otherwise} \end{cases}$$

$$(5)$$

Thus, the probability $P_{\text{source}}(x_i)$ for each $x_i \in D$ directly corresponds to the normalized attention weight $\mathbf{P}_{m:n}$, while the probability for vocabulary token not in $D$ is 0.

### 3.2 LargePiG: Vocabulary Distribution

The generation of the vocabulary distribution in the LargePiG model is seamlessly integrated with the output of the original LLM. This integration is achieved through the model's final component, an affine transformation layer commonly called the classification layer. This layer maps the output of the last Transformer layer $H_N$, to the vocabulary distribution $P_{\text{vocab}}$ over the vocabulary set $\mathcal{V}$. The probability distribution for the next token $x_t$ given the preceding sequence $x_{<t}$, is computed by applying a softmax function to the affine-transformed output:

$$P_{\text{vocab}}(x_t) = q_N(x_t \mid x_{<t}) = \text{softmax}\left(\phi\left(h_{t-1}^{(N)}\right)\right)_{x_t}, \quad x_t \in \mathcal{V}$$

$$(6)$$

where $h_{t-1}^{(N)}$ is the output vector from the last Transformer layer for the position $(t-1)$ in $H_N$, and $\phi(\cdot)$ performs the affine transformation to project this vector into the vocabulary space. The subscript $x_t$ indicates that we extract the probability corresponding to the token $x_t$ from the softmax output. This approach ensures that the generative capabilities of the underlying LLM are preserved within our LargePiG framework. Through this methodology, LargePiG leverages the extensive linguistic and syntactic knowledge of the LLM, thereby significantly retaining the richness and fluency of the generated query.

## 3.3 LargePiG: Copy Probability

The copy probability in our LargePiG model leverages the difference between the vocabulary distribution of the LLM's high layers and the last layer. For the layer $j$, we also compute the vocabulary distribution using $\phi(\cdot)$ as follows, where $\mathcal{J}$ is a set of candidate layers and this operation is called early exiting [41, 44]:

$$q_j\left(x_t \mid x_{<t}\right) = \text{softmax}\left(\phi\left(h_{t-1}^{(j)}\right)\right)_{x_t}, \quad j \in \mathcal{J}. \tag{7}$$

Based on the findings of Chuang et al. [10] and early exit decoding research [13, 41], when LLMs generate function words (e.g., auxiliary verbs, prepositions, conjunctions), the vocabulary distribution $q_j\left(x_t \mid x_{<t}\right)$ stabilizes at high layers. In contrast, when generating factual knowledge words (e.g., names, places, dates), the vocabulary distribution continues to evolve at high layers. In the query generation task, we expect the factual content in the generated query primarily comes from the source document, while syntax and other forms are organized by the LLM. This implies we can use the vocabulary distribution $q_N\left(x_t \mid x_{<t}\right)$ from the last transformer layer as an anchor layer, and by calculating the distributional differences with the vocabulary distributions from other high layers, determining whether LLM is generating factual knowledge words or function words. A larger distributional difference suggests a higher likelihood of generating factual knowledge words. Since our goal is to ensure that the factual content of the generated query mainly comes from the input document, the copy probability should be higher in such cases, and vice versa. Therefore, the copy probability can be calculated as follows:

$$p_{\text{cp}} = O_{j \in \mathcal{J}} d\left(q_N\left(x_t \mid x_{<t}\right), q_j\left(x_t \mid x_{<t}\right)\right), \tag{8}$$

where $O$ can be an average $\frac{1}{|\mathcal{J}|}\sum$, a max, or a min operation, $d(\cdot, \cdot)$ is a distributional distance measure such as Jensen-Shannon Divergence [10, 32], and $\mathcal{J}$ is the set of high-layers around the anchor layer. We can control the intensity of copying by adjusting $O$ and $\mathcal{J}$. A larger range of $\mathcal{J}$ and $O$ being max increases the likelihood of copying, while a smaller range of $\mathcal{J}$ and $O$ being min decreases it.

The final distribution generated by LargePiG is given by:

$$P_{\text{LargePiG}}(x_t) = p_{\text{cp}} P_{\text{source}}(x_t) + (1 - p_{\text{cp}}) P_{\text{vocab}}(x_t). \tag{9}$$

## 3.4 Exploring the Internal Mechanisms of LargePiG

The key to LargePiG's functionality lies in LLM's ability to correctly reflect the current generated token's attention weights towards

the source document and generate factual knowledge words and function words in the pattern we mentioned in § 3.3.

**Regarding the pointer attention distribution**, we analyzed the causes of hallucinations in query generation in § 1, concluding that the attention modules in LLMs are more 'truthful' than the FFN modules and classification layer. The **factuality hallucination** mainly arises from the LLM's insufficient knowledge about the source document. Some studies have shown that knowledge is mainly stored in the FFN module of the transformer layer in pre-trained language model [11]. Even if the self-attention module correctly focuses on the relevant token, the FFN module may still produce factuality hallucinations due to insufficient pre-training [31]. Moreover, Jiang et al. [20] found that MLP modules have a more significant impact on incorrect outputs than attention modules, indicating that in the transformer layers of LLMs, attention modules are more 'truthful' than FFN modules. The **relevance hallucination** can be attributed to the softmax bottleneck issue inherent in LLMs [7, 51], where the model predicts the probability of each word across the entire vocabulary, struggling to differentiate between words that are almost equally likely in a given pre-training context but have different meanings in the current situation. The softmax bottleneck primarily stems from the final classification layer, which is structurally unrelated to the attention module in the transformer layer we use. In Appendix A.2, we further experimentally verify that the attention modules in LLMs are more 'truthful' than the FFN modules and the classification layer.

**Regarding the copy probability**, we delve deeper into the findings of [10, 41], questioning why LLM predictions for function words stabilize at high layers' vocabulary distributions, while predictions for factual knowledge words do not. Research on early exit decoding [13, 41, 44] has demonstrated that different data samples (tasks) possess varying complexities. For multi-layer stacked deep models, such as ResNet [16] and LLaMA [45], simple tasks may only require shallow layers for completion, whereas complex tasks demand the involvement of all layers. The scaling law [22] and the emergence ability [49] also testify to this, with the model's ability to solve more complex tasks increasing alongside its size and layer number. Returning to our task, predicting function words can exit at shallower layers, while predicting factual knowledge words requires deeper layers, indicating that predicting function words is simpler, whereas predicting factual knowledge is more complex.

Why is predicting function words simpler, and predicting factual knowledge more complex? Achille et al. [1] demonstrated that tasks with greater information content are more complex. Since LLMs learn from human language, if we can verify that factual knowledge words in human language convey more information than function words, then the pattern mentioned above is determined by the nature of human language itself. Our experimental analysis within our TruthfulVQG and TruthfulDQG benchmarks investigated the semantic impact of removing factual knowledge words versus function words, with experimental details provided in Appendix A.3. The results show that on both datasets, removing factual knowledge words causes a greater decrease in semantic similarity scores with the original sentence compared to function words. These findings confirm that factual knowledge words contribute more significantly to the sentence's informational content

 

**Table 1: Performance comparisons between LargePiG and the baselines. The boldface represents the best performance. '†' means improvements are significant (paired t-test at $p$-value $< 0.05$).**

| Model | Qwen1.5 7B Chat | | | | | | LLaMA2 7B Chat | | | | | |
|---|---|---|---|---|---|---|---|---|---|---|---|---|
| | TruthfulVQG | | | TruthfulDQG | | | TruthfulVQG | | | TruthfulDQG | | |
| | MC1 | MC2 | MC3 | MC1 | MC2 | MC3 | MC1 | MC2 | MC3 | MC1 | MC2 | MC3 |
| **Base** | 40.35 | 66.97 | 37.70 | 27.34 | 85.77 | 39.83 | 52.94 | 75.12 | 46.01 | 33.72 | **71.61** | 34.29 |
| + CD | 35.79 | 63.43 | 36.49 | 24.25 | 85.60 | 37.99 | – | – | – | – | – | – |
| + DoLa | 37.97 | 64.73 | 35.68 | 23.52 | 85.05 | 37.09 | 52.79 | 75.25 | 46.10 | 35.09 | 69.97 | 33.19 |
| + LargePiG | **41.49**$^\dagger$ | **68.12**$^\dagger$ | **38.92**$^\dagger$ | **29.91**$^\dagger$ | **89.33**$^\dagger$ | **42.18**$^\dagger$ | **54.56**$^\dagger$ | **76.15**$^\dagger$ | **47.20**$^\dagger$ | **37.23**$^\dagger$ | 70.95 | **36.93**$^\dagger$ |
| **PQGR** | 43.61 | 70.08 | 41.26 | 25.86 | 77.23 | 36.86 | 52.22 | 74.21 | 45.60 | 32.28 | 65.74 | 31.41 |
| + CD | 41.71 | 66.10 | 40.69 | 23.84 | 77.90 | 35.58 | – | – | – | – | – | – |
| + DoLa | 40.13 | 66.50 | 38.24 | 23.79 | 76.51 | 35.67 | 51.83 | 73.69 | 44.54 | 31.92 | 64.41 | 31.52 |
| + LargePiG | **45.52**$^\dagger$ | **70.79**$^\dagger$ | **42.54**$^\dagger$ | **27.12**$^\dagger$ | **79.20**$^\dagger$ | **38.35**$^\dagger$ | **52.87**$^\dagger$ | **74.87**$^\dagger$ | **46.27**$^\dagger$ | **34.66**$^\dagger$ | **68.34**$^\dagger$ | **34.21**$^\dagger$ |
| **InPars** | 44.35 | 70.77 | 41.56 | 26.09 | 78.82 | 37.37 | 52.53 | 74.53 | 45.85 | 30.66 | 64.43 | 30.32 |
| + CD | 43.91 | 68.90 | 39.82 | 24.06 | 77.20 | 35.69 | – | – | – | – | – | – |
| + DoLa | 40.35 | 66.90 | 38.48 | 24.48 | 77.57 | 36.96 | 51.59 | 74.33 | 44.86 | 29.87 | 63.97 | 29.52 |
| + LargePiG | **46.26**$^\dagger$ | **71.51**$^\dagger$ | **42.82**$^\dagger$ | **27.34**$^\dagger$ | **81.17**$^\dagger$ | **38.53**$^\dagger$ | **53.03**$^\dagger$ | 74.74 | **46.20**$^\dagger$ | **33.70**$^\dagger$ | **67.30**$^\dagger$ | **33.36**$^\dagger$ |
| **AQG** | 40.50 | 67.26 | 37.85 | 27.41 | 85.86 | 39.93 | 54.00 | 75.92 | 46.87 | 34.82 | **71.62** | 34.42 |
| + CD | 36.79 | 63.36 | 33.44 | 24.23 | 83.56 | 37.96 | – | – | – | – | – | – |
| + DoLa | 37.99 | 64.65 | 35.62 | 25.59 | 85.28 | 39.21 | 52.79 | 75.25 | 46.10 | 33.02 | 70.96 | 33.17 |
| + LargePiG | **41.56**$^\dagger$ | **68.13**$^\dagger$ | **39.06**$^\dagger$ | **29.99**$^\dagger$ | **89.58**$^\dagger$ | **42.35**$^\dagger$ | **54.84**$^\dagger$ | **76.73**$^\dagger$ | **47.76**$^\dagger$ | **37.09**$^\dagger$ | 71.04 | **36.82**$^\dagger$ |

than function words, highlighting the complexity of predicting factual knowledge words. Verifying that the pattern found in [10, 41], rooted in the linguistic properties of human language, is a principle that holds true across multiple languages, even though initial studies focused on English scenarios. Our subsequent experiments expanded this understanding to multiple languages, validating the feasibility of employing this pattern for calculating copy probability in LargePiG. For further analysis of the effectiveness of copy probability in LargePiG, see Appendix A.3.

## 4 EXPERIMENT

### 4.1 Experimental Settings

**Datasets.** To quantitatively assess the truthful query generation capabilities of LargePiG in both video (e.g., TikTok) and document (e.g., Bing Search) scenarios, considering the absence of relevant datasets, we constructed two challenging benchmarks named TruthfulVQG and TruthfulDQG. These benchmarks correspond to formats similar to TruthfulQA [29], crafted from video (Chinese corpus) and document (English corpus) respectively, to validate the model's query generation truthfulness. The construction of the benchmarks utilized a combination of LLM and manual methods. The completed data format is shown in Table 8 of Appendix A.5, where "Bad queries" are those containing either relevance hallucinations or factuality hallucinations or both, "Good queries" are those without any hallucinations, and "Best query" represents the optimal query. The construction process is detailed in Appendix A.4 and

Appendix A.5, and the statistical results of the datasets are shown in Table 9.

**Metrics.** To evaluate LLMs in truthful query generation, we independently compute each reference query's log-probability. Drawing inspiration from the evaluation metrics of TruthfulQA-MC [10, 29], the metrics used to assess the truthfulness of the model-generated queries include MC1 (the percentage of all data where the best query log-probability is greater than all bad queries log-probability), MC2 (normalized total probability assigned to the set of good queries), and MC3 (the percentage of all good queries where each good query log-probability is greater than all bad queries log-probability).

**Models and Baselines.** We employed two types of backbone LLMs, Qwen1.5 7B chat [3] and LLaMA2 7B chat [45], and utilized four LLM-based query generation approaches, including (1) **Base**: using the backbone LLMs to directly generate queries in a zero-shot manner; (2) **PQGR** [12]: prompting the LLM with 8 in-context examples to generate queries, which achieves more suitable queries compared to the Base approach; (3) **Inpars** [5]: includes not only good queries in the in-context examples but also bad queries to enable the model to generate better queries through comparison; (4) **AQG** [25]: employ LoRA [17] to fine-tuning the LLM using real-world user-input queries and context data to enhance the model's query generation capability. The implementation details of these LLM-based query generation approaches are in Appendix A.6. Our approach, **LargePiG**, is model-agnostic and can be applied to different LLM-based query-generation methods, reducing the relevance and factuality hallucinations associated with model-generated queries. The implementation details of LargePiG

Table 2: Experimental results on multimodal data.

| Model | MC1 | MC2 | MC3 |
|---|---|---|---|
| **LLaVA-7B** | 58.40 | 80.45 | 51.54 |
| + LargePiG | **59.80** | **81.74** | **52.94** |
| **LLaVA-13B** | 57.20 | 79.18 | 50.74 |
| + LargePiG | **58.10** | **79.93** | **51.26** |

are provided in Appendix A.7. For baseline models, we compared LargePiG with recent closely related work aimed at reducing hallucinations in LLMs: **DoLa** [10], which enhances factuality in LLMs by decoding through contrasting layers, and Contrastive Decoding (**CD**) [27], which improves factuality in LLMs' generations by leveraging the contrasts between LLMs of different sizes, selecting tokens that maximize their log-likelihood difference. For Qwen1.5 7B chat, we chose Qwen1.5 1.8B chat [3] as the contrast model for CD. Since there is no smaller-sized model for LLaMA2 7B chat, we could not perform CD experiments on this model. *DoLa, CD, and LargePiG are all training-free decoding methods for reducing hallucinations in LLM generation, making them fair for comparison.*

## 4.2 Results

**Main result.** As shown in Table 1, LargePiG has demonstrated improvements across two datasets, various backbone methods, and different metrics, validating LargePiG's ability to enhance the truthfulness of LLM-based query generation methods. The effectiveness observed across datasets in different languages further corroborates the analysis presented in Section 3.4. Moreover, our method has surpassed CD and DoLa, which even exhibited negative gains on some datasets. The primary reason is that query generation primarily relies on the factual knowledge in the inputs, requiring less generated factual knowledge from the model, whereas DoLa and CD stimulate the model's knowledge by contrasting shallow layers' logits with deep layers' logits or contrasting large LLM's logits with small LLM's logits, which may lead to the generation of facts that do not align with the context. In the following analysis experiments, we will further discuss the respective advantages of CD, DoLa and LargePiG, and analyze in detail from the perspectives of relevance hallucinations and factuality hallucinations how LargePiG can improve the truthfulness of LLM generation. Further verification in Appendix A.13 confirmed that queries generated by LargePiG exhibit higher similarity to the real queries. Additionally, human evaluation and case studies in Appendix A.13 showed that LargePiG not only reduced relevance and factual hallucinations in the generated queries but also made them more appealing to users.

**Multimodal result.** LargePiG is effective not only on large language models but can also be applied to Large Vision-Language Models (LVLM), further enhancing the truthfulness of query generation that integrates both vision and language modalities. We selected the recently popular large vision-language model LLaVA [24] as the backbone model. Detailed method descriptions about the implementation can be found in Appendix A.8. To validate LargePiG's ability to address hallucination issues in multimodal query generation tasks, we compiled a multimodal version of the TruthfulVQG

Table 3: Experiment results on FACTOR.

| Model | LLaMA-7B | | LLaMA-13B | |
|---|---|---|---|---|
| | News | Wiki | News | Wiki |
| Base | 58.3 | 58.6 | 61.1 | 62.6 |
| + CD [26] | - | - | 62.3 | 64.4 |
| + DoLa [10] | 62.0 | 62.2 | 62.5 | 66.2 |
| + LargePiG | **71.0** | 60.4 | **72.1** | 63.1 |
| + DoLa + LargePiG | 63.4 | **64.7** | 65.3 | **68.8** |

dataset, named TruthfulVQG-M. Experimental results on LLaVA-7B/13B, shown in Table 2, indicate that the truthfulness of queries generated by LargePiG surpasses those produced by the original decoding method, confirming the effectiveness of LargePiG in multimodal tasks. We also observed that LLaVA-13B performs less effectively than LLaVA-7B, a potential reason being that in the video query generation task, due to the high noise level in video content, the more complex LLaVA-13B model might be more sensitive to noise. Furthermore, short videos contain some new content not present in the pre-training data, which could lead to easier overfitting to the training data in a zero-shot scenario, thus resulting in suboptimal performance compared to LLaVA-7B.

## 4.3 Analysis

**LargePiG's ability to reduce factuality hallucinations.** To specifically validate LargePiG's capability to address factual hallucinations, we selected the News and Wiki categories of FACTOR dataset [33], which assesses LLMs' factuality in long-paragraph settings by completion task. The News' ground-truth answers are based on facts from news content, which LLMs may not have sufficiently learned during training; the Wiki contains general facts well-learned during pre-training, allowing LLMs to respond based on pre-trained knowledge and also to learn from the context. To ensure a fair comparison with DoLa, we chose LLaMA-7B and LLaMA-13B as the backbone LLMs following DoLa's setting.

The experimental results shown in Table 3 demonstrate that on the News dataset, LargePiG successfully enhanced the copy ability of Base models to address hallucinations, thereby significantly outperforming other methods that solely rely on the model's intrinsic pre-trained knowledge and original context understanding capabilities. Given the feature of the Wiki dataset, although the results for LargePiG on Wiki do not surpass other methods that stimulate the model's own pre-trained knowledge, they still exceed the base model, validating the contribution of LargePiG's copy ability to resolving hallucinations. Moreover, LargePiG can be combined with state-of-the-art methods that are based on the model's pre-trained knowledge, achieving advancements beyond the current state of the art (i.e., +DoLa + LargePiG > +DoLa). This suggests that LargePiG's copy ability can be synergistically integrated with the model's inherent pre-trained knowledge.

**LargePiG's ability to reduce relevance hallucinations.** To independently verify LargePiG's capability to resolve relevance hallucinations, we generated queries using different models and

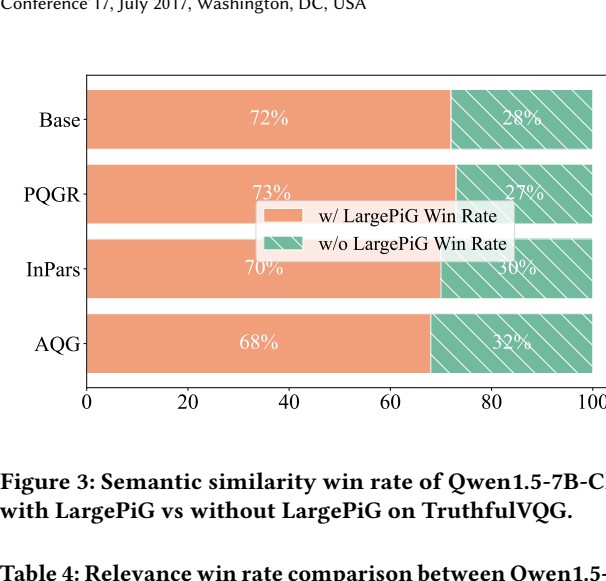

**Figure 3: Semantic similarity win rate of Qwen1.5-7B-Chat with LargePiG vs without LargePiG on TruthfulVQG.**

**Table 4: Relevance win rate comparison between Qwen1.5-7B-Chat with LargePiG and without LargePiG on TruthfulVQG.**

| Model | LargePiG Win | Original Model Win | Tie |
|-------|--------------|--------------------|----|
| **Base** | 827 | 70 | 103 |
| **PQGR** | 749 | 181 | 70 |
| **InPars** | 805 | 141 | 54 |
| **AQG** | 831 | 73 | 96 |

then encoded them and the corresponding context using the current state-of-the-art text representation model BGE [50] to calculate their cosine semantic similarity. The pairwise comparisons of cosine similarity are presented on Figure 3, demonstrating that LargePiG notably outperforms the baseline models. The results on TruthfulDQG are detailed in Appendix A.9, which presents similar conclusions to those found in the experiments on TruthfulVQG. This indicates that LargePiG effectively reduces the relevance hallucinations of query generation. In addition, we used GPT-4o (from OpenAI) to assess LargePiG's ability to reduce relevance hallucinations (see prompt in Appendix A.11). Considering time and API cost factors, we sampled 1000 data points from TruthfulVQG for evaluation. The experiments in Table 4, judged by GPT-4o, further confirm that LargePiG can mitigate relevance hallucinations.

**LargePiG's ability to copy.** To validate whether LargePiG has a stronger copy ability compared to the original LLM decoder, we tested the performance of LLM with and without the addition of LargePiG on tasks that require copying from the inputs. Following the setting of Jelassi et al. [18] for validating LLMs' copy capability, we selected the SQuAD question-answering dataset [38], which provides text paragraphs along with several questions pertaining to the text and features various inputs lengths. We conducted experiments on Qwen1.5-7B-Chat, reported the $F_1$ score, and classified questions into short and long categories based on whether their length exceeded 200 words. The results on Figure 4 show that LargePiG significantly improved the $F_1$ score on Qwen1.5-7B-Chat, with more pronounced improvements for scenarios with long inputs, indicating that LargePiG indeed enhances the copy ability of LLMs. Similar results on LLaMA2-7B-Chat are shown in Appendix A.10.

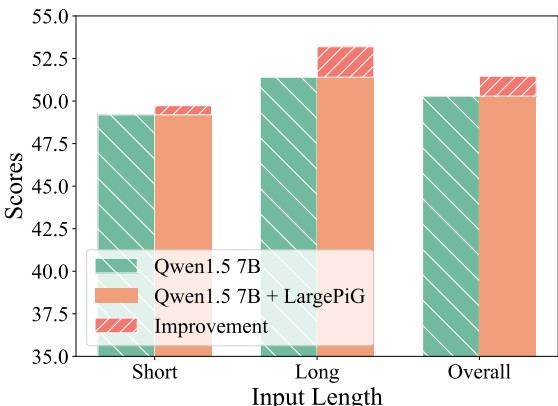

**Figure 4: Comparison of the Copying Ability between Qwen1.5-7B-Chat and Qwen1.5-7B-Chat with LargePiG on the SQuAD dataset.**

**Table 5: Decoding latency (ms/token).**

| | Baseline | DoLa | LargePiG |
|--|----------|------|----------|
| **Base / AQG** | 95.9 (×1.00) | 99.9 (×1.04) | 101.8 (×1.06) |
| **InPars** | 135.1 (×1.00) | 142.4 (×1.05) | 139.8 (×1.03) |
| **PQGR** | 142.0 (×1.00) | 148.3 (×1.04) | 149.1 (×1.05) |

**Efficiency analysis.** We use NVIDIA V100-32G GPUs and 52-core Intel(R) Xeon(R) Gold 6230R CPUs at 2.10GHz machine to analyze the efficiency of original decoding (baseline), DoLa, and LargePiG when applied across different query generation models. The decoding time of LargePiG in LLaMA2-7B models increases by a maximum of 6% compared to the baseline and is on par with the decoding time of DoLa, as shown in Table 5 (experiments on Qwen1.5-7B are detailed in Appendix A.12). The results demonstrate that LargePiG can enhance the truthfulness of query generation with negligible additional time consumption, proving the practical applicability of LargePiG.

## 5 CONCLUSIONS

LLM-based query generation significantly improves query quality and user experience in information retrieval systems, but it also introduces hallucination challenges, hindering its application in emerging use cases such as "related search". To address these, we propose LargePiG, a training-free method transforming an LLM into a Pointer-Generator. LargePiG separates content and form in LLM-generated queries, using input knowledge for fact generation and LLM capabilities for syntactic structure. It combines self-attention weights for pointer attention distribution, LLM original output as vocabulary distribution, and high-layer vocabulary distribution for copy probability. Our empirical evaluations on the proposed TruthfulVQG and TruthfulDQG datasets confirm LargePiG's effectiveness in reducing hallucination on query generation tasks.

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

## A APPENDIX / SUPPLEMENTAL MATERIAL

### A.1 Experimental Verification of Hallucination Classification

Our hallucination classification for generated query is grounded in real-world observations, aiming to help readers better understand the distinct types of hallucinations present in query generation. This categorization is intended to offer valuable insights for future research in this domain. To further validate our classification, we conducted an analysis experiment using the TruthfulVQG dataset (Detailed in Section 4.1). In this experiment, we encoded both the generated queries and corresponding video content using BGE [50] and computed the cosine similarity to obtain a **Semantic Similarity score**. The results demonstrate that factual hallucinations can occur independently, even in the absence of relevance hallucinations, highlighting the need to decouple these two types of hallucinations for more precise handling.

The experimental results in Table 6 confirm that factual hallucinations can persist even with high semantic similarity scores. This finding underscores the importance of treating relevance and factual hallucinations separately to improve query generation and retrieval quality.

### A.2 Evaluation of attention modules are more 'truthful' than FFN modules

In section 3.4, we propose that Attention modules are more 'truthful' than other modules in LLMs (e.g., FFN modules). To validate this core observation, we conducted experiments on the RAGTruth dataset, which is a word-level hallucination corpus in various tasks within the Retrieval-augmented generation (RAG). The RAGTruth dataset contains responses from various LLMs that exhibit hallucinations in RAG scenarios, with manually annotated hallucination spans, hallucination types, and hallucination reasons. The RAG scenario is similar to the query generation scenario, as both involve generation based on input content, approximating the query generation context.

Specifically, in our validation experiments, we selected data from RAGTruth [34] where LLaMA2-7B-Chat exhibited hallucinations. Using LangChain [2], a widely used open-source toolkit, we applied the RecursiveCharacterTextSplitter to segment the input retrieved document into different spans. We then calculated whether the attention module attended span (mean pooling the attention scores then selecting the input span with the highest score) of LLaMA2-7B-Chat during the generation of hallucination spans could identify the hallucination spans in the response. This evaluation was based on GPT4-o (from OpenAI) using the following prompt:

```
Prompt: { external  context + query}

Respond: {response}

Conflict  Span: { Conflict  Span}
Conflict  Type: { Conflict  Type}
Reason: {Reason}

Given  the  following  context  information：  "{Attend Span
    ↪  }",  can  this  support  the  existence  of  a
    ↪   conflict  in  the  response?  Please  answer with "
    ↪  Yes"  or  "No"  and  give  the  reason  on  the
    ↪  newline.
```

The results are shown in Table 7. We found that, in most cases, the attention modules of LLMs can attend to the correct input spans to identify hallucinations in the response. This indicates that hallucinations in LLMs are caused by other modules in LLMs (e.g., FFN modules), thereby proving the core observation that Attention modules are more 'truthful' than other modules in LLMs.

### A.3 Implementation Details of Words Information

In the experiments concerning word information, we conducted tests using the TruthfulVQG and TruthfulDQG benchmarks constructed in this paper. For English in the TruthfulDQG benchmark, we used Spacy [3] for tokenization and part-of-speech tagging, while for TruthfulVQG (Chinese corpus), we employed Jieba [4] and Hanlp [5]. Factual knowledge words include organizations, personal names, locations, and dates. Function words include auxiliary verbs, prepositions, determiners, conjunctions, and coordinating conjunctions. Subsequently, on both datasets, we removed an equal number of factual knowledge words and function words and then utilized BGE embeddings [50] to align and compare the cosine similarity

---

[2] https://www.langchain.com/.
[3] https://spacy.io
[4] https://github.com/fxsjy/jieba
[5] https://www.hanlp.com

**Table 6: Semantic similarity scores across query types, highlighting that factual hallucinations can occur despite high similarity with relevant content.**

| Type | Max Semantic Similarity | Min Semantic Similarity | Average Semantic Similarity |
|---|---|---|---|
| Facticity hallucination queries | 0.8492 | 0.3792 | 0.6479 |
| Facticity truth queries | 0.8607 | 0.2647 | 0.6482 |
| Random similarity | N/A | N/A | 0.2709 |

**Table 7: Proportion of data where Llama2 7B Chat attention heads attend to the correct information.**

| Attention heads attend | Attention heads mis-attend |
|---|---|
| 77.5% | 22.5% |

between the modified sentences and the original sentences. The results are shown below:

- In the TruthfulDQG benchmark, removing factual knowledge words resulted in a similarity score of 0.7741, while removing function words led to a higher similarity score of 0.9296.
- In the TruthfulVQG benchmark, the removal of factual knowledge words produced a similarity score of 0.7415, compared to 0.9477 when function words were eliminated.

The results show that on both datasets, removing factual knowledge words causes a greater decrease in semantic similarity scores with the original sentence compared to function words. These findings confirm that factual knowledge words contribute more significantly to the sentence's informational content than function words, highlighting the complexity of predicting factual knowledge words. Verifying that the pattern found in [10, 41], rooted in the linguistic properties of human language, is a principle that holds true across multiple languages, even though initial studies focused on English scenarios.

**Why Can LLM Identify Factual Knowledge Words and Function Words?**

Considering that LLMs can only directly learn to predict the next word in the natural language training corpus, they may not have an intuitive concept of what constitutes factual knowledge words and function words. Therefore, we conducted an intrinsic frequency analysis of factual knowledge and function words on the TruthfulDQG benchmark. The statistical results are shown below:

- Number of different words in function words: 228
- Number of different words in factual knowledge words: 3263
- Total number of words in function words: 33849
- Total number of words in factual knowledge words: 6026
- Average occurrence of function words: 148.46
- Average occurrence of factual knowledge words: 1.85

These results show that function words appear much more frequently than factual knowledge words, particularly evident from their average occurrences. It is evident that due to the substantially larger training data of function words compared to factual knowledge words, LLMs can predict function words at shallower layers while predicting factual knowledge words need deeper layers.

**Another Perspective on the Effectiveness of Copy Probability in LargePiG.**

Besides the pattern we mentioned above, Jiang et al. [20] observes that in hallucinated cases, the output token's information rarely shows abrupt increases and maintains consistent superiority in high layers of the LLMs. This corresponds to cases in LargePiG where there is a higher copy probability, thus enabling the reduction of hallucinations by copying factual knowledge words from the source document. This further demonstrates the capability of the copy probability in LargePiG to address the issue of hallucinations.

## A.4 Details about Dataset Collection

The TruthfulVQG dataset is collected from a real short video platform used by over one billion users. The TruthfulDQG dataset is adapted from the MS-MARCO dataset [4]. The data processing for TruthfulVQG is more complex than TruthfulDQG's. Thus, we will use TruthfulVQG as an example to illustrate the process.

**Data Collection:**

The raw data was collected from Search Click Data and Post-Watch Search Data, and the final processed public data does not include any user search information, only video content, and LLM-generated queries.

- **Collected Data Source:**
  - **Search Click Data (30,000 samples):** We collect 30,000 samples of users' clicked videos after searching the corresponding queries with data flowing from query to video.
  - **Post-Watch Search Data (10,000 samples):** We collect 10,000 samples of users' searched queries after watching the corresponding videos, which is a smaller subset compared to click data, with data flowing from video to query.
- **Criteria for Inclusion:**
  - **Search Click Data:** Include only data with positions greater than one and less than twenty to mitigate position bias of the top results and low relevance of farther results.
  - **Post-Watch Search Data:** Include only data with total count numbers greater than five to ensure relevance to previously viewed videos.

**Components of Video Content:**

- **Title:** Accurate representation of video content.
- **Video Dialogue Text (ASR):** Prone to noise but contain detailed information about the video.
- **Video Text Information (OCR):** More reliable than ASR and contains more information than Title.

**Data Preprocessing:** Remove examples lacking textual features, containing sensitive words, or background music that affect ASR results.

Next, we will use LLMs to generate multiple queries for data annotation of all videos. To enable the LLMs have the ability to generate high-quality queries, we first fine-tuned these LLMs. Then, we combined them with the original LLMs to generate queries.

**Model Fine-Tuning:**

- **Models Used:**
  - **Qwen1.5 7B Chat** [3] and **InternLM 7B Chat** [6] [6]: Among the strongest for Chinese language capabilities.
- **Purpose:**
  - Employing multiple LLMs ensures diversity in generated queries, reducing the risk of repetitive queries that single model sampling might produce.

**Data Utilization and Query Generation**

- Sort data by video quality scores and select the top 10,000 samples for query generation (Generation is time-intensive, approximately 40 hours per week. Hence, only the top entries are used).
- Approximately 20+ queries are generated per video using the following prompt.

**Query Generation Prompt:**

```
instruction : Based on the video's title, dialog text, and
    ↪ text information within the video, generate a
    ↪ relevant and engaging search query. This query
    ↪ should accurately reflect the video content,
    ↪ adhere to factual information, and stimulate user
    ↪ interest to drive clicks. Ensure the query is
    ↪ concise and contains key information points.
input: Title : { Title content}
       Dialog text: {Dialog text content}
       Text information: {Text information content}
       Query:
output: {Query content}
```

This prompt is also used in our experiments to generate queries [7].

## A.5 Details about Dataset Annotation.

During the data annotation section, we first performed further cleaning and filtering of the data. We utilized a combination of LLM and manual annotation to label TruthfulDQG and TruthfulVQG. This hybrid approach of LLM and manual annotations has been employed in numerous works on hallucination benchmark annotation [8, 30].

*A.5.1   Phase One: Filter Dataset.* Remove sensitive words and perform heuristic query quality filtering based on repetitiveness and length scores.

*A.5.2   Phase Two: Relevance Assessment.* This phase focuses on detecting relevance hallucination by measuring the relevance of generated queries to the video content.

**Similarity Calculations**

(1) **Embedding-Based Similarity:** Utilizes BAAI BGE Embedding [50] and cosine similarity to compute similarity scores between text embeddings.
(2) **Word-Based Similarity:** Employs Jieba for text segmentation and calculates similarity using the Jaccard similarity [8].

**Weighting Method** Adjusts relevance scoring based on the ASR noise level:

$$\text{ASR Score} = 0.6 \times \cos(\text{ASR}, \text{OCR}) + 0.4 \times \cos(\text{ASR}, \text{Title})$$

$$\text{Query Scoring} = \begin{cases} 0.34 \times (\text{Query}, \text{Title}) + \\ \quad 0.33 \times (\text{Query}, \text{ASR}) + \\ \quad 0.33 \times (\text{Query}, \text{OCR}), & \text{if ASR Score} > 0.5 \\ 0.4 \times (\text{Query}, \text{Title}) + \\ \quad 0.2 \times (\text{Query}, \text{ASR}) + \\ \quad 0.4 \times (\text{Query}, \text{OCR}), & \text{if ASR Score} > 0.3 \\ 0.5 \times (\text{Query}, \text{Title}) + \\ \quad 0.1 \times (\text{Query}, \text{ASR}) + \\ \quad 0.4 \times (\text{Query}, \text{OCR}), & \text{otherwise} \end{cases}$$

*A.5.3   Phase Three: Factuality Assessment.* Detecting the factuality hallucination of the generated queries by using LLM-based fact-checking methods–Self-Check (4-shot CoT) and FacTool [9].

**Self-Check (4-shot CoT).** We implement Self-Check (4-shot CoT) using the larger and more powerful LLM Qwen1.5-72B-Chat [3] to detect queries' factuality hallucination. The prompt is shown below [9]:

```
You will receive a query generated by another model. Your
    ↪ task is to check whether this query contains any
    ↪ factual errors. Please refer to the examples and
    ↪ guidelines below when evaluating the query:
- If the query accurately reflects verifiable facts, it
    ↪ should be considered factually correct.
- If the query contains misleading or inaccurate
    ↪ information, it should be considered factually
    ↪ incorrect.
- If you cannot determine the accuracy of the query, or
    ↪ if the query requires more context for
    ↪ evaluation, it should be considered
    ↪ indeterminate.
- Your response must follow the specified format,
    ↪ containing two keys: "reasoning" (the process
    ↪ of reasoning) and "factuality" (the judgment
    ↪ of factuality, where True if the query is
    ↪ factually correct or does not involve factual
    ↪ information; False if the query contains
    ↪ factual errors; No if indeterminate).
You must respond only in the format described below.
    ↪ Do not reply in any other form. Adding any
    ↪ content that violates the response format is
    ↪ prohibited. Start your response with '{{'.
```

---

[6] We replace InternLM 7B Chat with LLaMA2 7B Chat on TruthDQG.

[7] As the TruthfulVQG is a Chinese Dataset, we translate the prompt from Chinese using ChatGPT-4.

[8] https://scikit-learn.org/stable/modules/generated/sklearn.metrics.jaccard_score.html

[9] As the TruthfulVQG is a Chinese Dataset, we translate the prompt from Chinese using ChatGPT-4.

```
[Response Template]:
{{
  "reasoning": "Reason whether the query is  factual .
      ↪ Think through step by step .",
  " factuality ": "True if the query is  factually  correct
      ↪  or does not involve  factual information ;
      ↪ False  if the query contains  factual  errors ;
      ↪ No if indeterminate ."
}}

Examples:
1. [Query]: " Collapse of a tunnel in Antarctica "
{{
  "reasoning": "This query contains a factual  error .
      ↪ Given the extremely low temperatures in
      ↪ Antarctica ,  constructing  tunnels is
      ↪ extremely  difficult , and based on current
      ↪ knowledge, there are no tunnels in
      ↪ Antarctica ,  thus a  collapse cannot occur .",
  " factuality ": False
}}

2. [Query]: "The Asian Games in Hangzhou will open on
      ↪ September 23, 2023"
{{
  "reasoning": "The factuality  of this query cannot be
      ↪ determined with the information at hand; it
      ↪ requires  consultation of the latest  official
      ↪  announcements or news sources to  verify  the
      ↪  specific  opening date .",
  " factuality ": No
}}

3. [Query]: "How to make scrambled eggs with tomatoes"
{{
  "reasoning": "This query is  not about the
      ↪  truthfulness of a statement but requests a
      ↪ recipe ,  therefore  it does not involve
      ↪ factual  errors .",
  " factuality ": True
}}

4. [Query]: "Messi is  Argentine"
{{
  "reasoning": "This query is  factually  correct . Lionel
      ↪  Messi is a well–known football player born
      ↪ in Argentina, a fact  that is widely known
      ↪ and can be  verified  through reliable  sources
      ↪  .",
  " factuality ": True
}}

Below is the given query –
[Query]: {}
```

**Advanced Fact-Checking.** For indeterminate cases after Self-Check, we use advanced fact-checking tools FacTool [9] with Qwen1.5

72B Chat [3] and Serper [10] to further check queries' factuality based on external data sources from Google Search. The prompt is shown below [11]:

```
You are an excellent  assistant .
You will  receive a piece of text . Your task is to
    ↪ identify any factual  errors within this text .
When judging the  factuality  of the given text , you may
    ↪ refer to  provided  evidences  if  necessary .
These evidences could be  helpful . Some evidences might
    ↪  contradict each other . You must be
 careful  when using evidences to  assess the  factuality
    ↪  of the given text .
The response should be a dictionary  containing  three
    ↪ keys – "reasoning ", " factuality ",
" error ", and " correction ", corresponding to the
    ↪ reasoning ,  whether the given  text  is
true (Boolean value – True or False ), the  factual  error
    ↪  present in the  text , and the
corrected  text .
Below is the given text
[ text ]: {query}
Below is the  provided  evidence
[evidences ]: {evidence}
You should respond only in the format described below.
    ↪ Do not return any other  content .
Start  your response with   '{{'.
[response format]:
{{
  "reasoning ": "Why is the given  text  factual  or not?
      ↪ Be  careful  when you claim
 something is  not  factual . When you claim something is
      ↪  not  factual ,  you must provide
 multiple  pieces  of  evidence to  support your decision
      ↪  .",
  " error ": " If  the  text  is  factual , then None;
      ↪ otherwise ,  describe  the error .",
  " correction ": " If  there  is an error , then the
      ↪ corrected  text .",
  " factuality ": " If  the given  text  is  factual , then
      ↪ True; otherwise ,  False ."
}}
```

Finally, the completed data format is shown in Table 8, and the statistics of TruthfulVQG and TruthfulDQG are shown in Table 9.

**Human Assessment.** To further ensure the relevance and factual accuracy of the query, we request three annotators with graduate-level qualifications to manually evaluate the "good queries" to confirm factuality and relevance to the context, ensuring they are both engaging and appropriate.

---

[10] The website of Serper is https://serper.dev/.
[11] As the TruthfulVQG is a Chinese Dataset, we translate the prompt from Chinese using ChatGPT-4.

**Table 8: Description of data fields in TruthfulVQG and TruthfulDQG.**

| | Video / Document Content | Best query | Good Queries | Bad Queries |
|---|---|---|---|---|
| **Data Type** | string | string | [string] | [string] |
| **Description** | Description of the video / document content | Best query (factual and most relevent) | Array of good queries | Array of bad queries |

**Table 9: Statistics of TruthfulVQG and TruthfulDQG. # denotes the average number.**

| Dataset | Data Count | # Good Queries | # Bad Queries | # Total Queries | Language |
|---|---|---|---|---|---|
| **TruthfulVQG** | 4,148 | 3.82 | 4.75 | 8.56 | Chinese |
| **TruthfulDQG** | 2,718 | 4.04 | 4.00 | 8.05 | English |

## A.6 Implementation Details of LLM-based Query Generation Approaches

The prompts used on TruthfulDQG for different LLM-based query generation approaches are shown below (The prompts used on TruthfulVQG are just different in the instruction, which has been demonstrated on Appendix A.4):

**Base / AQG:**

```
Given the following document, generate a concise, factual
    ↪ and relevant query that a user might type into a
    ↪ search engine to find this information.
Document: {Document contents}.
Related Query:
```

**PQGR:**

```
Given the following document, generate a concise, factual
    ↪ and relevant query that a user might type into a
    ↪ search engine to find this information.
Example 1:
Document: {Document contents}.
Related Query: {The query relevant and factual to document
    ↪   contents}.
...
Example 9:
Document: {Document contents}.
Related Query:
```

**InPars:**

```
Given the following document, generate a concise, factual
    ↪ and relevant query that a user might type into a
    ↪ search engine to find this information.
Example 1:
Document: {Document contents}.
Related Query: {The query relevant and factual to document
    ↪   contents}.
Hallucination Query: {The query irrelevant and unfactual to
    ↪   document contents}.
...
Example 4:
Document: {Document contents}.
Related Query:
```

The size of the dataset for LoRA fine-tuning AQG is 10,000 pairs. The fine-tuning targets the q_proj and v_proj within the transformer layers. The learning rate is set to 5e-5, the per-device train batch size is 4, and the gradient accumulation steps are 4.

## A.7 Implementation Details of LargePiG.

We run all the experiments on machines equipped with NVIDIA V100 GPUs and 52-core Intel(R) Xeon(R) Gold 6230R CPUs at 2.10GHz. We utilize the Huggingface Transformers package to conduct experiments. During the decoding of responses from the language models, we employ random sampling with a temperature of 0.8 and a maximum of 256 new tokens to generate responses. The rest of the parameters use the models' default settings. As for selecting the layer to calculate the pointer attention distribution, we used the last layer's attention weights by comparing them with other layers. As for selecting the words to calculate the pointer attention distribution, we recommend filtering the function words in the input using tools detailed in Appendix A.3. Considering that the Jensen-Shannon divergence is usually small in the high-dimensional space of vocabulary distribution, we scale the copy probability $p_{cp}$ in LargePiG by a factor of $\alpha$. To ensure that the scaled $p_{cp}$ remains within a reasonable range, we clip its value to be less than 0.5, thus maintaining a balance between copy and generation. The value of $\alpha$ is selected from the set [100, 500, 1000]. The $O_{j \in \mathcal{J}}$ in Equation 8 is selected as $\max_{j \in \mathcal{J}}$, and $\mathcal{J}$ comprises the last 8 or 16 layers of the backbone LLMs, excluding the anchor layer which is the last layer (for increased efficiency, either even or odd numbered layers may be selected). We use two-fold validation to select the hyper-parameters. The LLaMA2-7B-Chat can be downloaded from https://huggingface.co/meta-llama/Llama-2-7b-chat-hf. The Qwen1.5-7B-Chat can be downloaded from https://huggingface.co/Qwen/Qwen1.5-7B-Chat. Due to the limited Chinese training corpus of LLaMA2-7B-Chat, we used Llama2-Chinese-7b-Chat on TruthfulVQG, which can be downloaded from https://huggingface.co/LinkSoul/Chinese-Llama-2-7b.

## A.8 Details about LargePiG Applied to LLaVA

The architecture of LLaVA [24] is straightforward, comprising only a Vision Encoder, Projection, and Language Model, with training

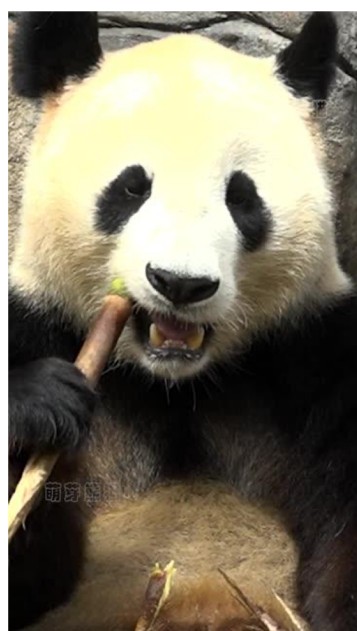

(a) Video cover one. Map tokens: Cat, gray, black, elve, eyes, sitting ...

(b) Video cover two. Map tokens: pandas, white, fang, gry, Chinese ...

**Figure 5: An example of two video covers mapped to tokens, where we have ignored other irrelevant words and the "_" character before some tokens.**

conducted in two stages: Stage 1: Pre-training for Feature Alignment, and Stage 2: Fine-tuning End-to-End. A key issue when applying LargePiG to LLaVA concerns how to map image tokens to text tokens, thus establishing an attention distribution based on the source content. Considering during the Feature Alignment stage, the primary task is aligning the image features $\mathbf{H}_v$ with the pre-trained LLM word embeddings, we propose mapping each image token to the closest text token in the embedding space when computing the Pointer Attention Distribution. In the implementation, we utilize the faiss vector database [21] to store text token embeddings and retrieve the corresponding tokens using image token embeddings, allowing for rapid retrieval of relevant tokens. Case studies shown in Figure 5 reveal that this retrieval method can accurately reveal the main information in the images, although many noise tokens are also retrieved. Therefore, we apply rule-based filtering to remove tokens with low similarity to the text part and construct the attention distribution using the remaining tokens together with the text tokens.

## A.9 More results on LargePiG's Ability to Reduce Relevance Hallucinations

More results on LargePiG's ability to reduce relevance hallucinations are shown in Figure 6 and the left of Figure 7, both LLaMA2-7B-Chat and Qwen1.5-7B-Chat with LargePiG can generate more semantic relevance queries with the document / video contents, indicating that LargePiG is effective in reducing the relevance hallucinations of query generation.

## A.10 More Results on LargePiG's Copy Ability

The results of LargePiG with LLaMA2-7B-Chat on the SQuAD question-answering dataset are shown on the right of Figure 7, which also show that LargePiG significantly improved the $F_1$ score on LLaMA2-7B-Chat, with more pronounced improvements for scenarios with long inputs, indicating that LargePiG indeed enhances the copy ability of LLMs.

## A.11 Prompt for relevance evaluation by GPT4-o

The evaluation prompt are shown below:

```
You will be given a description of a video and queries
    ↪ generated by the baseline model and the LargePiG
    ↪ model based on the video description . Your task is
    ↪  to determine which model generates higher–quality
    ↪  queries . When evaluating the queries , please
    ↪ refer to the following guidelines :

1. Are the queries  relevant to the video description ?
2. You must reply only in the format described below.
    ↪ Adding any extra content that violates the reply
    ↪ format is  prohibited .

Begin your reply with  '{{.

[Reply Template]:
```

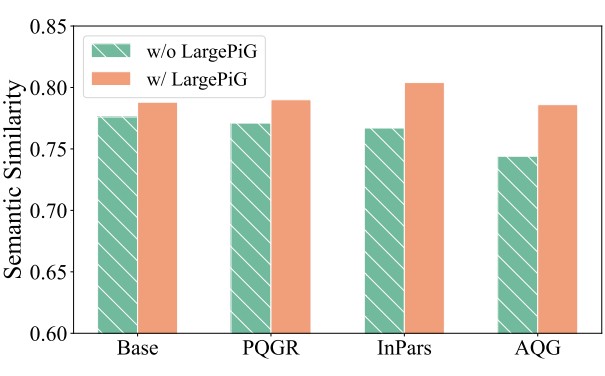 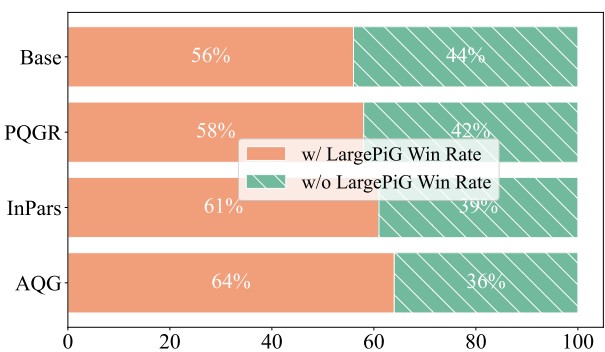

**Figure 6: Results of LLaMA2-7B-Chat without LargePiG vs with LargePiG on TruthfulDQG. Left: Overall semantic similarity scores. Right: Win rate with LargePiG compared against without LargePiG.**

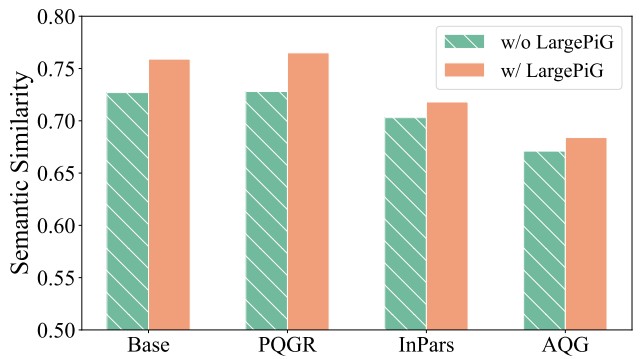 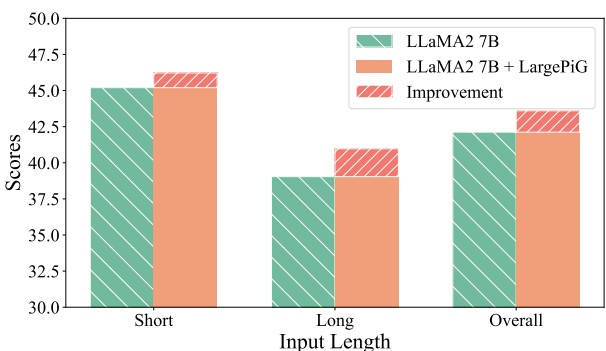

**Figure 7: Left: Overall semantic similarity scores of Qwen1.5-7B-Chat with LargePiG vs without LargePiG on TruthfulVQG. Right: Performance of LLaMA2-7B-Chat vs LLaMA2-7B-Chat + LargePiG on SQuAD.**

```
{{
'win_model': 'LargePiG (LargePiG generated query is more
    ↪ relevant) or Baseline (Baseline generated query is
    ↪ more relevant) or Tie (both models generated
    ↪ similar queries)',
'reason': 'The reason for determining the winning model in
    ↪ the previous statement'
}}

Video description: `{}`

Baseline generated query: `{}`

LargePiG generated query: `{}`
```

## A.12 More Results on Efficiency Analysis

Table 10 shows the decoding latency for different models on Qwen1.5-7B. It is evident that compared to LLaMA2-7B, the inference latency of Qwen1.5-7B is significantly reduced. Therefore, the addition of DoLa or LargePiG, although increasing the time cost compared

to LLaMA2-7B, still shows a relatively small overall increase. The maximum increase in time cost is about 10%, which is within an acceptable range.

## A.13 Generated Query Quality Evaluation

To verify the quality of queries generated by LargePiG compared with the baseline models, we first encode the generated queries and the corresponding user-input queries using BGE Embedding. Subsequently, we compute the cosine similarity to compare the semantic similarity between queries generated by different models and those input by users. As can be observed from Table 11 and 12, queries generated by LargePiG exhibit higher similarity to actual user input queries, thereby confirming the high quality of LargePiG-generated queries from a semantic relevance perspective.

To further validate the performance of LargePiG in real-world scenarios, we compared it with the online query generation model (A 7 billion parameter transformer decoder-only model) of a short-video platform with billions of users, observing the performance of the online model after integrating LargePiG. In the offline evaluation, we retrieved the results generated by the online model along with the corresponding input data (totaling 1108 samples) and

**Table 10: Decoding latency (ms/token) on Qwen1.5-7B.**

|  | Baseline | DoLa | LargePiG |
|---|---|---|---|
| **Base / AQG** | 64.21 (×1.00) | 70.67 (×1.10) | 68.56 (×1.07) |
| **InPars** | 69.54 (×1.00) | 75.82 (×1.09) | 76.50 (×1.10) |
| **PQGR** | 82.26 (×1.00) | 92.45 (×1.12) | 89.49 (×1.09) |

**Table 11: Semantic Evaluation of Generated Query Quality on the Qwen1.5 7B Chat.**

|  | Qwen1.5 7B Chat | | | | Qwen1.5 7B Chat + LargePiG | | | |
|---|---|---|---|---|---|---|---|---|
|  | **Base** | **PQGR** | **InPars** | **AGQ** | **Base** | **PQGR** | **InPars** | **AGQ** |
| **TruthfulDQG** | 0.6751 | 0.6629 | 0.6586 | 0.6858 | 0.7086 | 0.6893 | 0.6725 | 0.7077 |
| **TruthfulVQG** | 0.6143 | 0.5983 | 0.5957 | 0.6102 | 0.6260 | 0.6074 | 0.6056 | 0.6247 |

**Table 12: Semantic Evaluation of Generated Query Quality on the LLaMA2 7B Chat.**

|  | LLaMA2-7B-Chat | | | | LLaMA2-7B-Chat + LargePiG | | | |
|---|---|---|---|---|---|---|---|---|
|  | **Base** | **PQGR** | **InPars** | **AGQ** | **Base** | **PQGR** | **InPars** | **AGQ** |
| **TruthfulDQG** | 0.6574 | 0.6561 | 0.6561 | 0.6568 | 0.6850 | 0.6897 | 0.6979 | 0.6870 |
| **TruthfulVQG** | 0.5898 | 0.5687 | 0.5674 | 0.5898 | 0.6003 | 0.5792 | 0.5875 | 0.5953 |

**Table 13: Human and LLM evaluations of the queries generated by LargePiG and the original online model.**

|  | Baseline Win | LargePiG Win | Tie |
|---|---|---|---|
| Count Number | 54 | 1031 | 23 |

conducted generation after adding LargePiG to the online model. Subsequently, we employed a collaborative approach of LLM and human evaluation. Initially, the Qwen1.5-72B-Chat was used to determine whether LargePiG wins, the base model wins, or if it is a tie, providing reasons for each. Then, two human evaluators with graduate-level qualifications reviewed the LLM's outputs, correcting any erroneous assessments made by the LLM, thereby enhancing the overall efficiency and accuracy of the evaluations. Combined with experimental results on Table 13 and case studies (translated from Chinese) below, it was demonstrated that adding LargePiG not only reduced the relevance and factual hallucinations in the generated queries but also made them more attractive to users, further validating the effectiveness of LargePiG. From the analysis of case studies, we found that the reason why LargerPiG can generate queries that are more attractive to users may be the interpolation of vocabulary distribution, which can reduce the probability of generating an end token. Moreover, during the query generation process, there is a consistent high alignment with the video content. Consequently, the generated queries are more detailed and specific, thereby attracting more user clicks.

**Evaluation Prompt:**

```
You will  receive a video's description , along with queries
    ↪ generated by the  baseline  model and the  LargePiG
    ↪ model based on that  video  description . Your task
    ↪ is  to  determine  which model produced the  higher
    ↪ quality  query. When evaluating the  queries ,  please
    ↪  refer  to  the  following  guidelines :
        – Whether the query is  relevant to  the  video
            ↪  description
        – Whether the query is  factually  accurate
        – Whether the query can  attract  user  interest
        – You must reply only  in  the  format described
            ↪ below. Do not respond in any other
            ↪ form. Adding any extra  content  that
            ↪  violates  the  reply  format  is
            ↪ prohibited . Start your reply with
            ↪  '{{'.
[Reply  Template]:
{{
    "win_model": "LargePiG ( if  LargePiG generated a
        ↪  better  query) or Baseline ( if
        ↪ Baseline generated a  better  query) or
        ↪ Tie ( if  both models generated  similar
        ↪ queries) ",
    "reason": "The reason  for  the  previous winning
        ↪ model decision"
}}
Video  description :  {}
```

Query generated by  baseline :  {}
Query generated by LargePiG:  {}

**Case studies:**

Example One:

video_content : Video Category: Film and Television . Video
↪ Tags: Mermaid. Video Title :  The most  terrifying
↪ hunters  often  appear  in  the  guise  of  their  prey ,
↪ horror   thriller  mermaid  series  #FastTalkingFilm  #
↪ KuaishouCinemaSuperstarPlan #FilmCommentary.
↪ Video Cover: Fish Cost. Video Author:  Firebird  Films
↪ ,
Baseline :
1. Mermaid Movie,
2. Mermaid Movie Commentary,
3. Mermaid Movie Episode One,
4. Commentary on the First  Episode  of  Mermaid,
5. Complete Mermaid Movie Online Viewing,
6.  First  Part  of  Mermaid,
7. Second Part  of  Mermaid Movie,
8. Complete Mermaid Movie Online Viewing Uncut,
LargePiG:
1. Mermaid Movie,
2. Mermaid Movie Online Viewing,
3. Complete Mermaid Video,
4. Horror Mermaid Movie,
5. High  Definition   Online  Viewing  of  Mermaid Movie,
6. Free  High  Definition   Online  Viewing  of  Horror
↪ Mermaid Movie,
7. Uncut Super High  Definition   Online  Viewing  of
↪ Mermaid Movie,
8. Mermaid Movie Commentary,
Judge  Result :
win_model: LargePiG,
reason : The LargePiG's  queries  contain  more  specific
↪ information ,  such  as  'horror movie' and ' free
↪ high  definition  online  viewing ,' which more
↪ accurately   reflect  the  video's  theme (horror
↪ thriller )  and  are  likely  to  generate  more  user
↪ interest .  Additionally ,  it  offers  a  variety  of
↪ viewing options ,  such  as  the  uncut  super  high
↪ definition   version ,  which  may  be  more
↪ appealing  to  users .

Example Two:

video_content : Video category :  Film  and TV show; Video tags
↪ :  Farewell  Song; Video  title :  Xiao  Zhe  argues  with
↪ An Duo, An Duo proposes divorce,  and they  are
↪ destined  to  break up! #CatchTheNewDrama #
↪ WebDramaFarewellSong; Video cover: Attending an
↪ international music festival ;  Video creator :  Old
↪ Friend Qi ( recruiting   apprentices ),
Baseline :
1. Farewell  Song TV series ,
2. Farewell  Song episode 36  finale ,
3. Farewell  Song An Duo and Xiao Zhe,

4. Farewell  Song TV series  online  viewing,
5. An Duo and Xiao Zhe,
6. An Duo Xiao Zhe,
7. Where to watch Farewell  Song TV series ,
8. Farewell  Song  finale ,
9. An Duo Xiao Zhe arguing.
LargePiG:
1. Farewell  Song TV series ,
2. Farewell  Song An Duo and Xiao Zhe,
3. An Duo and Xiao Zhe arguing,
4. An Duo and Xiao Zhe divorce ,
5. An Duo Farewell Song,
6. Xiao  Zhe An Duo,
7. Xiao  Zhe Farewell  Song An Duo finale ,
8. Xiao  Zhe An Duo divorce truth .
Judge  Result :
win_model: LargePiG,
reason : The LargePiG model generated queries  are  more
↪ specific ,  incorporating  key events  from the
↪ video  description  (arguing ,  divorce )  which
↪ better   reflect  the  video  content  and  engage
↪ user  interest ,  such  as  'An Duo and Xiao Zhe
↪ arguing',  'An Duo and Xiao Zhe divorce ',  'Xiao
↪ Zhe An Duo divorce truth '.  In  contrast ,  the
↪ Baseline  queries ,  although  relevant ,  do  not
↪ sufficiently   highlight  the  conflicts  and  plot
↪ developments in  the  video ,  such  as  ' finale '
↪ and 'online  viewing ',  which  are  more  general
↪ and  factually    incorrect .

Example Three:

video_content : Video Category: Anime Video Tags: Ultraman
↪ Video Title :  Episode 4 | Go Ahead, Ultraman Zero!
↪ Performed by: Shota Minami / voyager Lyrics  by:
↪ Hideki Tamiya/Koki Konishi Composed by: Tomohiro
↪ Yamaguchi Edited by: Ultraman Information  Master
↪ Ultraman Golden Songs Zero Edition  #Ultraman #
↪ ZeroUltraman 15th Anniversary Video Cover: Go
↪ Ahead Ultraman Zero Performance by: voyager /  Shota
↪ Minami Composed by: Tomohiro Yamaguchi Lyrics by:
↪ Hideki Tamiya/Koki Konishi Edited  by:  Ultraman,
↪ Information  Master Video Author: Ultraman
↪ Information  Master,
Baseline :
Go Ahead, Ultraman Zero,
Ultraman Zero,
Go Ahead, Ultraman Zero Song,
Complete Lyrics  of  Ultraman Zero Theme Song,
Go Ahead, Zero,
Original  Singer  of  Go Ahead, Ultraman Zero,
How to Sing Go Ahead, Ultraman Zero Song,
Original  Sound  of  Go Ahead, Zero,
Ultraman Zero Go Ahead,
LargePiG:
Go Ahead, Ultraman Zero,
Original  Singer  of  Go Ahead, Ultraman Zero Song,

Go Ahead, Ultraman Zero Theme Song Lyrics,
Go Ahead, Ultraman Zero Anime Episode One,
Go Ahead, Ultraman Zero Theme Song,
Go Ahead, Ultraman Zero Ultraman Zero Song,
Ultraman Zero,
How to Sing Go Ahead, Ultraman Zero Song.,
Ultraman Zero Go Ahead,
Original Singer of Go Ahead, Ultraman Zero Ultraman
↪ Zero Song.,
Judge Result:

win_model: LargePiG,
reason: LargePiG's queries are more specific,
↪ containing more information related to the
↪ video content such as 'Go Ahead, Ultraman Zero
↪ Anime Episode One', which can stimulate user
↪ interest and provide a richer background
↪ related to the video. In contrast, Baseline's
↪ queries, while related to the video theme, are
↪ more generic and do not specify details such
↪ as the original singer or anime episodes.

