# OpenReview forum: "LargePiG for Hallucination-Free Query Generation: Your Large Language Model is Secretly a Pointer Generator"
_ACM.org/TheWebConf/2025/Conference — WWW 2025 Poster_

### Official Review · Reviewer_jkN1 · 2024-11-25

**Novelty:** 5
**Technical Quality:** 5

**Review:**

This paper introduces LargePiG, a novel model designed to achieve hallucination-free query generation by transforming Large Language Models (LLMs) into Pointer Generators. LargePiG employs a model-agnostic and training-free approach, leveraging the LLM’s inherent attention weights for pointer attention distribution and calculating copy probability based on the differences between vocabulary distributions from higher layers and the final layer of the model. Experimental results on two datasets encompassing document and video scenarios demonstrate the superiority of LargePiG in reducing hallucinations. Additionally, LargePiG improves the accuracy of document-based question-answering and factuality evaluation tasks in large vision-language models. The source code and datasets are made available for further research.

**Strengths:**

* The author innovatively introduced "relevance hallucination" and "factuality hallucination" as new classifications, providing a deeper understanding of the hallucination issues in query generation using LLMs.
* The proposed LargePiG method successfully reduces hallucinations by decoupling content and form through Pointer Generator. Its model-agnostic and training-free nature offers flexibility and ease of integration with various LLMs.
* Extensive experiments conducted on two datasets covering document and video scenarios validate the effectiveness of LargePiG.
* The source code and datasets provided by authors facilitates further exploration by the research community.

**Weaknesses:**

* In the experimental section, especially regarding multimodal data experiments, more baseline models should be incorporated [1,2].
* While experiments on efficiency are provided, additional theoretical analysis of computational complexity would help readers assess the method's scaling capabilities.
* The authors claim that "LargePiG causes negligible latency in the query generation process" and support this with experiments in Table 5. However, this conclusion is one-sided. In real-world web serving scenarios, acceleration techniques like Flash Attention [3] and Paged Attention [4] are common practice to avoid inefficient decoding. The compatibility with these techniques largely determines a method's practicality in real web serving scenarios. However, LargePiG requires access to attention weights of MHA, which appears incompatible with these techniques. Moreover, it's unclear whether baseline models in Table 5 used Flash Attention, making the comparison less meaningful for real-world applications. The authors should actively supplement discussions on this to ensure LargePiG's effectiveness in practical web application scenarios.

**Questions:**

1. Can methods like DoLa [2] be applied to multimodal data in the same way as LargePiG?
2. Under what implementation were the efficiency experiments conducted? Were inference systems like vllm [4] or SGLang [5] used?


[1] Huang, Qidong, et al. "Opera: Alleviating hallucination in multi-modal large language models via over-trust penalty and retrospection-allocation." *Proceedings of the IEEE/CVF Conference on Computer Vision and Pattern Recognition*. 2024.

[2] Chuang, Yung-Sung, et al. "Dola: Decoding by contrasting layers improves factuality in large language models." arXiv preprint arXiv:2309.03883 (2023).

[3] Dao, Tri, et al. "Flashattention: Fast and memory-efficient exact attention with io-awareness." *Advances in Neural Information Processing Systems* 35 (2022): 16344-16359.

[4] Kwon, Woosuk, et al. "Efficient memory management for large language model serving with pagedattention." *Proceedings of the 29th Symposium on Operating Systems Principles*. 2023.

[5] https://github.com/sgl-project/sglang

**Reviewer Confidence:**

4: The reviewer is certain that the evaluation is correct and very familiar with the relevant literature

**Scope:**

4: The work is relevant to the Web and to the track, and is of broad interest to the community

---

### Official Review · Reviewer_boKf · 2024-11-27

**Novelty:** 5
**Technical Quality:** 4

**Review:**

### Summary
This paper proposes a method for Query Generation using LLM as a Pointer Generator. The insight is that when autoregressively generating factual knowledge, the model should preferentially select words from the source document (a segment of words in the input text), while for generating function words, the model should rely more on its own internal generation. Specifically, LargePiG leverages the inherent attention mechanisms of LLMs to separate content from form, ensuring that the factual knowledge from the input is preserved while the syntactic structure is accurately compiled. This is achieved through a model-agnostic and training-free decoding method, which transforms the LLM into a Pointer-Generator. The effectiveness of LargePiG is validated through extensive experiments on two datasets, TruthfulVQG and TruthfulDQG, covering both document and video scenarios. The results demonstrate that LargePiG significantly reduces hallucinations while maintaining or improving the efficiency of query generation.

### Pros
- Model-Agnostic and Training-Free: LargePiG can be applied to any LLM without requiring architectural modifications or additional training, making it highly versatile and easy to deploy.
- Reduction in Hallucinations: The method effectively reduces both relevance and factuality hallucinations, enhancing the reliability of generated queries.
- Efficiency: LargePiG maintains or improves the efficiency of query generation, as evidenced by the experimental results.
- Decoupling Content and Form: By separating the content (factual knowledge) from the form (syntactic structure), LargePiG ensures that the generated queries are both accurate and well-formed.
- Comprehensive Evaluation: The authors constructed two challenging benchmarks, TruthfulVQG and TruthfulDQG, to thoroughly evaluate the performance of LargePiG.

### Cons
- Insufficient Sensitivity Analysis of Key Parameters: The paper defines the copy probability $p_{cp}$ and mentions its dependence on the operation $\mathcal{O}$, the set of high layers $\mathcal{J}$, and the factor $\alpha$, but lacks a thorough sensitivity analysis of these key parameters. Understanding how changes in these parameters affect $p_{cp}$ and the overall performance of LargePiG is crucial for optimizing the model and ensuring its robustness across different tasks and datasets.
- Scalability and Real-Time Performance: The paper does not address the scalability of LargePiG. Further investigation is needed to assess its performance on extremely long texts and its ability to achieve real-time query generation without significant performance degradation.
- Comparative Analysis with Other Methods: While the paper mentions DoLa and CD as baseline methods, a more comprehensive comparison with other recent approaches for hallucination mitigation in LLMs, such as data-driven methods or retrieval-augmented generation, would strengthen the paper's contribution and provide a clearer understanding of LargePiG's advantages and limitations.

**Questions:**

- Q1: The copy probability $p_{cp}$ is defined as $p_{cp} = \mathcal O_{j \in \mathcal{J}}$ $d(q_N(x_t|x_{<t}), q_j(x_t|x_{<t}))$, where $\mathcal{O}$ is an operation (e.g., average, max, or min) and $\mathcal{J}$ is a set of high layers around the anchor layer. Additionally, the paper mentions in Appendix A.7 that $p_{cp}$ is clipped to 0.5 with a factor $\alpha$. Could the authors provide a more detailed analysis of how $p_{cp}$ varies under different strategies for $\mathcal{O}$ and $\mathcal{J}$, as well as the impact of $\alpha$ on the performance of LargePiG?
- Q2: Can LargePiG handle extremely long input texts without significant performance degradation and achieve real-time query generation?
- Q3: Could you provide a more detailed comparison of LargePiG with other recent methods for hallucination mitigation in LLMs, such as data-driven approaches or retrieval-augmented generation?

**Reviewer Confidence:**

3: The reviewer is confident but not certain that the evaluation is correct

**Scope:**

4: The work is relevant to the Web and to the track, and is of broad interest to the community

---

### Official Review · Reviewer_mbLP · 2024-11-30

**Novelty:** 6
**Technical Quality:** 6

**Review:**

The work presented in the sources introduces LargePiG, a novel method for reducing hallucinations in LLM-generated queries.

 Quality:
 1.Strong empirical evidence: The sources provide extensive experimental results on multiple datasets, including TruthfulVQG, TruthfulDQG, FACTOR, and SQuAD, demonstrating LargePiG's effectiveness in reducing both relevance and factuality hallucinations.
 2. Comparison with state-of-the-art: LargePiG is compared against other hallucination mitigation techniques like DoLa and Contrastive Decoding, showing consistent improvements.
 3. Ablation studies: The impact of different components of LargePiG, such as the copy probability and pointer attention distribution, are analyzed.

 Clarity:
 1. Well-defined concepts: The sources clearly define relevance and factuality hallucinations, distinguishing them as distinct problems.
 2. Detailed methodology: The three components of LargePiG – Pointer Attention Distribution, Vocabulary Distribution, and Copy Probability are explained in detail, with clear mathematical formulations.
 3. Illustrative examples: The use of real-world examples from short-video platforms helps illustrate the problem of query hallucinations and the potential benefits of LargePiG.

 Originality:
 1. Novel approach: Turning the LLM into a Pointer-Generator for query generation is a novel approach that addresses the specific challenges of this task.
 2. Training-free implementation: Unlike traditional Pointer-Generator models, LargePiG does not require any additional training, leveraging intrinsic features of the LLM.
 3. Leveraging LLM patterns: LargePiG's copy probability calculation exploits the distinct patterns observed in LLM generation of function words and factual knowledge words.

 Significance:
 1. Addresses a crucial problem: Hallucinations in LLM-generated queries can significantly impact user experience and hinder the development of new applications.
 2. Applicable to various LLMs: The model-agnostic nature of LargePiG makes it applicable to a wide range of LLMs, potentially benefiting various NLP tasks.
 3. Potential for multi-lingual applications: While the sources primarily focus on Chinese and English, they suggest that the underlying principles of LargePiG may extend to other languages.

 Pros:
 1. Effective in reducing hallucinations: Empirically demonstrated to outperform baseline models and other mitigation techniques.
 2. Preserves LLM capabilities: Does not require retraining or modifications to LLM architecture.
 3. Efficient and easy to implement: Leverages intrinsic LLM features, minimizing computational overhead.

 Cons:
 1. Reliance on LLM's internal mechanisms: The effectiveness of LargePiG depends on certain assumptions about the LLM's internal workings, which may not hold true for all LLMs.
 2. Requires further validation for multi-lingual applications: The generalizability of LargePiG's copy probability mechanism to other languages needs to be further investigated.

**Questions:**

While LargePiG has been shown to be effective for query generation, it's unclear how it would perform in other NLP tasks that suffer from hallucinations, such as summarization or dialogue generation.
Question 1: Have you explored applying LargePiG to other NLP tasks beyond query generation? If so, what are the results?
Question 2: What modifications or adjustments might be necessary to adapt LargePiG to different task-specific requirements?

**Reviewer Confidence:**

3: The reviewer is confident but not certain that the evaluation is correct

**Scope:**

4: The work is relevant to the Web and to the track, and is of broad interest to the community

---

### Official Review · Reviewer_Dqyd · 2024-12-02

**Novelty:** 4
**Technical Quality:** 3

**Review:**

### Summary
The submitted paper first motivates the need for truthfulness within the query generation task, based on examples from content-video or short-video platform applications. In particular, it distinguishes between factual hallucination and relevance hallucination.
The paper then proposes using pointer generators to mitigate the hallucination of LLMs. This approach is training-free and can be applied during inference by modifying the attention mechanism of the decoder component, without the need for further fine-tuning.
The paper introduces three new datasets for evaluating the truthfulness of generated queries: two based on textual data and one incorporating image multimodality.
Finally, the paper compares the proposed approach with other decoding strategies used to reduce hallucination (Contrastive decoding, DoLa).

### Strengths
* The paper proposes the novel application of pointer generators on LLMs to reduce hallucination in query generation, a method not previously used for this task.
* It introduces new datasets for evaluating the trustworthiness of query generation, including multimodal data.
* The approach can be applied to state-of-the-art models, such as PQGR or InPars, without requiring further fine-tuning.
* The evaluation includes human judgment to assess the effectiveness of the approach.
* The paper is well-written and easy to follow.

### Weaknesses
* While the paper introduces three new datasets for evaluation, it does not compare LargePiG on previously existing datasets, such as TruthfulQA (MC), TruthfulQA (Open-Ended Generation), Chain-of-Thought Reasoning on StrategyQA, or GSM8K.
* There is missing information about the proposed datasets (TruthfulDQG). Specifically, the dataset is created from MS MARCO, but it is unclear how the 2,718 samples were extracted from the hundreds of thousands of original queries in MS MARCO. This is particularly important as the paper proposes both a new dataset and a new model.
* The results for DoLa on the newly proposed datasets are inconsistent with its performance on existing datasets. While DoLa typically improves over the base LLM on prior datasets, it appears to decrease model performance on TruthfulVQG and TruthfulDQG (the proposed ones). This inconsistency may point to issues in the dataset creation process.
*  When attempting to reduce hallucination, there is often a trade-off between overall performance and truthfulness. This trade-off is not addressed in the paper, particularly regarding how much performance loss is observed with LargePiG on standard query generation datasets for ad-hoc retrieval tasks.

**Questions:**

See above.

**Reviewer Confidence:**

3: The reviewer is confident but not certain that the evaluation is correct

**Scope:**

3: The work is somewhat relevant to the Web and to the track, and is of narrow interest to a sub-community